# Effects of the Transfer Method and Interfacial Adhesion on the Frictional and Wear Resistance Properties of a Graphene-Coated Polymer

**DOI:** 10.3390/nano13040655

**Published:** 2023-02-08

**Authors:** Temesgen B. Yallew, Prashant Narute, Rakesh S. Sharbidre, Ji Cheol Byen, Jaesung Park, Seong-Gu Hong

**Affiliations:** 1Interdisciplinary Materials Measurement Institute, Korea Research Institute of Standards and Science, Daejeon 34113, Republic of Korea; 2School of Mechanical and Industrial Engineering, Mekelle University, Mekelle 231, Ethiopia; 3Physiological Signal Processing and Measurement Solution, Physionics, Daejeon 34027, Republic of Korea; 4Department of Nano Science, University of Science and Technology, Daejeon 34113, Republic of Korea

**Keywords:** graphene-coated polymer, frictional properties, wear resistance, graphene transfer method, graphene adhesion, nanoscratch test

## Abstract

Graphene is a promising candidate used to reduce friction and wear in micro- and nano-device applications owing to its superior mechanical robustness and intrinsic lubrication properties. Herein, we report the frictional and wear resistance properties of a graphene-coated polymer and how they are affected by fabrication processes. The results show that graphene deposited on a polymer substrate effectively improves both frictional and wear resistance properties, and the degree of improvement significantly depends on the graphene transfer method and interfacial adhesion between graphene and the substrate. Dry-transferred graphene showed better improvement than wet-transferred graphene, and the strong adhesion of graphene achieved by imidazole treatment aided the improvement. A combined analysis of surface morphology and scratch trace shows that the graphene transfer method and graphene adhesion dominate the structural integrity of the transferred graphene, and the graphene/substrate interfacial adhesion plays a decisive role in the improvement of both properties by suppressing the delamination of graphene from the substrate during the nanoscratch test, thereby preventing crack formation in graphene and weakening the puckering effect.

## 1. Introduction

Reducing friction- and wear-related mechanical failures in moving micro- and nano-device applications remains a challenging issue to address [1,2]. To date, considerable efforts have been devoted to this issue, and it has been discovered that friction and wear mechanisms vary with the materials comprising a device as well as the operating conditions such as environmental and tribological conditions [3,4]. Recently, as a potential remedy to resolve this issue, graphene (Gr) in the form of a thin layer (i.e., Gr coating) has received extensive attention due to its desirable combination of mechanical robustness, thermal conductivity, and tribological properties [5,6,7,8,9]. According to the results, Gr coated on various substrates, including polymers [10,11], aluminum alloys [12], and steels [4,13,14], can effectively improve their frictional and wear resistance properties. In addition, there are several factors affecting the improvement of both properties, such as Gr film thickness [15], structural defects of Gr [9], surface chemistry of Gr [16,17], and substrate properties [18]. Among these, the structural defects and surface chemistry of Gr are dominated by the fabrication process and are considered key factors. Despite their importance, however, systematic studies on this issue are still lacking.

In this study, we explored how Gr film fabrication processes, such as Gr transfer methods onto a substrate, affect the frictional and wear resistance properties of a Gr-coated polymer. Two well-established conventional Gr transfer methods, wet transfer and dry transfer, were employed, and an imidazole treatment process was further added to the Gr transfer processes to enhance Gr adhesion. The monolayer chemical vapor deposition (CVD)-grown Gr films were transferred to a currently used transparent polymer substrate, polyethylene terephthalate (PET), using the two transfer methods with or without imidazole treatment, and the fabricated Gr/PET films were scratched using a nanoscratch tester to measure their frictional and wear resistance properties. Analyses on surface morphology and scratch traces on the Gr/PET films were conducted using a scanning electron microscope (SEM) to understand the mechanisms improving their frictional and wear resistance properties.

## 2. Materials and Methods

### 2.1. Gr/PET Film Fabrication

Monolayer Gr was synthesized on a 35-μm-thick Cu foil (JX Nippon Mining and Metals Co., Tokyo, Japan) using a thermal CVD process. Details on the Gr synthesis process can be found in our previous works [19,20]. Next, Gr/PET films were fabricated using two well-established conventional Gr transfer methods: wet transfer [21] and dry transfer [22]. Figure 1 shows the schematic illustration of the two Gr transfer methods. For the wet transfer method (Figure 1a), first, a 5 wt% polymethyl methacrylate (PMMA) (CAS No. 9011-14-7, Sigma-Aldrich Co., Ltd., Seoul, Korea) solution was spin-coated onto the Gr/Cu foil; the Gr on the other side of the Cu foil was removed through O_2_ plasma treatment. Next, the PMMA/Gr/Cu structure was baked on a hot plate at 80 °C for 5 min and immersed in 0.1 M ammonium persulfate solution (APS) to remove the Cu foil. To improve graphene adhesion, imidazole-containing APS (0.1 M APS, 5 mM imidazole, and 50 mM sulfuric acid) was also used. The released PMMA/Gr structure was rinsed with deionized (DI) water for 30 min and picked up by a PET substrate. Finally, the PMMA layer was removed by immersion in acetone, and the Gr/PET film was rinsed in DI water several times. For the convenience of describing the experimental results, we denote the Gr/PET films fabricated using the wet transfer method without and with imidazole treatment as WT (APS) and WT (APS + IM) samples, respectively. In the dry transfer method (Figure 1b), first, a carrier film (CF) comprising a 40-μm-thick silicon adhesive and 112-μm-thick PET was mechanically attached to the CVD-grown Gr using a laminator (GMP Photoart-I7LSI, GMP Co., Ltd., Gyeonggi-do, Korea), and the Gr on the other side of the Cu foil was removed through O_2_ plasma treatment. Next, the CF/Gr/Cu structure was immersed in 0.1 M APS to remove the Cu foil, and the CF/Gr structure was rinsed with DI water for 30 min and then dried in a desiccator for 7 h. In the wet transfer method, imidazole-containing APS was also used to improve Gr adhesion. Next, the CF/Gr structure was laminated on a PET substrate using the laminator, and the CF was peeled off carefully, leaving a Gr/PET film. We denote the Gr/PET films fabricated using the dry transfer method without and with imidazole treatment as DT (APS) and DT (APS + IM) samples, respectively.

### 2.2. Structural Characterization

The microstructural properties of the Gr/PET films were characterized using a Raman spectrometer with a 532-nm laser as the excitation source (Alpha 300, WITec, Ulm, Germany). The measurements were made on an area of 50 × 50 μm^2^ with a 50× objective lens (0.55 numerical aperture), and the laser beam size was 2 μm.

Surface morphology, microstructural features, and scratch traces of the Gr/PET films were observed using field-emission SEM (Sigma 300 VP, Ziess Microscopy Ltd., Jena, Germany) operated at 1 kV [19].

### 2.3. Nanoscratch Test

Nanoscratch tests were conducted to measure the frictional and wear resistance properties of the Gr/PET films. A nanoscratch tester with a high-resolution cantilever and sphero-conical diamond-like carbon indenter (HR-309, NST^3^, CSM Instruments, Inc., Needham, MA, USA) was used, and the applied normal load (force) progressively increased during the scratch testing. The parameters used in the scratch testing were as follows: an initial applied normal load of 0.01 mN, final applied normal load of 2.0 mN, loading rate of 1.99 mN/min, scratch distance of 0.5 mm, scratch velocity of 0.5 mm/min, and indenter tip radius of 1µm. In the scratch tests, pre-scan and post-scan procedures were included to eliminate the effects of sample inclination and/or surface profile on the measurement.

## 3. Results and Discussion

### 3.1. Structural Properties

The SEM micrographs of the surface morphology of the fabricated Gr/PET films are shown in Figure 2, including the micrograph of a bare PET substrate for comparison; the cross-sectional SEM micrographs of the WT (APS + IM) and DT (APS + IM) samples are also presented in Appendix A. The quality (i.e., structural integrity) of the transferred Gr varied significantly with the Gr transfer method and imidazole treatment. Both the wet and dry transfer methods without the imidazole treatment caused notable structural damages, such as tear, crack, wrinkle, and fold, along with PMMA residues on the transferred Gr (Figure 2a,c). It is noted, however, that the dry-transferred Gr exhibited a somewhat better structural integrity than the wet-transferred Gr. The structural integrity of the transferred Gr was certainly improved by the imidazole treatment for both the wet and dry transfer methods (Figure 2b,d). Imidazole molecules are known to adsorb on Gr surfaces via π-π interactions, and these adsorbed molecules increase Gr’s surface energy, thereby enhancing the adhesion of Gr [19,22,23]. In the wet transfer method, the enhanced Gr adhesion results in improved interfacial adhesion between the transferred Gr and PET, preventing the wash out of the transferred Gr during the PMMA removal process using acetone. Hence, the structural damage in the transferred Gr is mitigated (Figure 2a,b). The interfacial adhesion of Gr with a target substrate is known to play a decisive role in the roll-based dry transfer process as the difference in the interfacial adhesion between Gr/CF and Gr/PET dominates the quality of the transferred graphene [23]. In this regard, the Gr adhesion enhanced by the imidazole treatment provides a strong interfacial adhesion between Gr and PET, thereby upgrading the quality of the transferred graphene (Figure 2c,d). As shown in Figure 2d, compared to other samples, the dry-transferred Gr with the imidazole treatment, DT (APS + IM) sample, exhibits a superior structural integrity without notable structural damages (untransferred Gr region, tear, crack, etc.) or PMMA residues.

Raman spectroscopy was used to characterize the transferred Gr. Figure 3 presents the Raman spectra of the Gr/PET films, where the data from the bare PET are inserted for comparison. For all the Gr/PET films, and although the Gr G band peak at ~1590 cm^−1^ overlaps with the strong PET background peak, the appearance of a 2D band peak at ~2680 cm^−1^ indicates the successful transfer of Gr onto the PET substrate [20,24]. In addition, the absence of a visible D band peak at ~1350 cm^−1^ indicates that the transferred Gr is of high quality, i.e., both the wet and dry transfer processes do not cause notable microstructural damage to Gr.

### 3.2. Frictional and Wear Resistance Properties

The frictional and wear resistance properties of the Gr/PET films were examined by measuring the friction coefficient and penetration depth during the nanoscratch tests. In the nanoscratch test, a normal force applied to the Gr/PET film progressively increased from 0.01 mN to 2 mN, and the later force required to move a scratch probe laterally across the Gr/PET film was measured. The nanoscratch test comprises a three-stage process [25,26]. First, the film surface is pre-scanned with a small normal force. This process does not cause any structural damage to the film surface and offers initial surface information of the film, such as the location of the scratch and the film surface roughness. Second, the nanoscratch test is carried out, where the scratch probe scratches the film surface and leaves a scratch trace on the film surface. Last, the scratched area is post-scanned with a sufficiently small load to obtain the film surface information after the nanoscratch test. The friction coefficient is defined as the ratio of the lateral force to the normal force during the nanoscratch test [25,26,27]. During this test, the Gr/PET film underneath the scratch probe experiences compressive stress, whereas tensile stress develops in the area around the edge of the scratch probe [28].

Figure 4 shows the comparison of the friction coefficient and penetration depth of the fabricated Gr/PET films, where the data from the bare PET substrate are inserted for comparison. The average values and standard deviation of 30 nanoscratch test results are plotted in Figure 4a,b (Appendix A), and these average values were used to calculate the reduction ratio of the friction coefficient and penetration depth with respect to the bare PET substrate, as shown in Figure 4c,d. As shown in Figure 4a,c, the Gr coating on the PET substrate clearly lowers the friction coefficient, thereby improving the frictional properties. Compared to the value of the bare PET, it is reduced by ~13% for WT (APS), ~25% for WT (APS + IM) and DT (APS), and ~40% for DT (APS + IM) (Figure 4c). Considering that graphene has a significantly lower friction coefficient (~0.03) [29] than PET (~0.3) [30,31], the reduction of the friction coefficient by the Gr coating is rationalized. It is noted that for all the Gr/PET films and the bare PET substrate, the friction coefficient gradually increases with the scratch distance (Figure 4a). This is because the normal force applied to the film progressively increased during the nanoscratch test, causing a deeper penetration of the scratch probe into the Gr/PET film, thereby leading to a different contact condition between the scratch probe and film with increasing scratch distance, and thus, requiring a greater lateral force for the scratch probe to scratch the film surface. Another interesting result was that the degree of improvement of frictional properties was significantly dependent on the Gr transfer method and imidazole treatment. Regardless of the imidazole treatment, the dry-transferred Gr exhibited a lower friction coefficient than the wet-transferred Gr; the reduction ratio of the friction coefficient was almost double in the dry-transferred Gr (Figure 4c). In addition, the imidazole treatment effectively reduced the friction coefficient for both the wet-transferred and dry-transferred Gr. After the imidazole treatment, the reduction ratio of the friction coefficient increased by ~100% for the wet-transferred Gr and ~60% for the dry-transferred Gr (Figure 4c).

The fabrication processes also affected the homogeneity of the frictional properties of the Gr/PET films. As shown in Figure 4a, there is a large scatter (i.e., a large standard deviation) in the friction coefficient data of the bare PET substrate, indicating the poor homogeneity of the PET substrate surface. However, the scatter was much reduced due to the Gr coating, indicating that the homogeneity of the PET substrate surface was improved. It is further noted that the reduction in scatter was more pronounced in the dry-transferred Gr than in the wet-transferred Gr, irrespective of the imidazole treatment. In addition, scatter was greatly reduced by the imidazole treatment in both the wet-transferred and dry-transferred Gr. These results imply that the Gr coating on the PET substrate effectively improves the structural homogeneity of the PET substrate surface, and the dry transfer method and imidazole treatment (i.e., enhancing the adhesion of Gr) are more effective for this purpose.

The variation in the penetration depth with the scratch distance is presented in Figure 4b,d. The Gr coating significantly reduced the penetration depth, improving the wear resistance performance; the reduction ratio of the penetration depth with respect to the bare PET substrate was ~36% for WT (APS), ~43% for WT (APS + IM), and >~50% for DT (APS) and DT (APS + IM). The improvement in the wear resistance performance was higher in the dry-transferred Gr than in the wet-transferred Gr, and the imidazole treatment further improved the wear resistance performance. It is noted that such characteristics in wear resistance are the opposite of those shown in frictional properties. This is because the low friction coefficient indicates the presence of the Gr coating on the PET substrate, and the excellent mechanical robustness of Gr prevents the scratch probe from penetrating the PET substrate, thereby reducing the penetration depth. On the contrary, the high friction coefficient implies that the Gr coating is structurally damaged, and thus the scratch probe readily penetrates the PET substrate, thereby increasing the penetration depth. In addition, the scatter of the penetration depth data was reduced by applying the Gr coating, dry transfer method, and imidazole treatment. This is attributed to the improved structural homogeneity of the Gr/PET film surface as in the friction coefficient.

Based on these results, we can infer that the Gr coating on the PET substrate effectively improves its frictional and wear resistance properties, and the degree of improvement is significantly affected by the fabrication process of the Gr/PET film. The dry transfer of Gr and the enhancement of Gr adhesion, achieved by the imidazole treatment, are advantageous for both properties.

### 3.3. Mechanisms Improving Frictional and Wear Resistance Properties

The SEM micrographs showing the scratch traces on the WT (APS + IM) and DT (APS + IM) samples are presented in Figure 5; the cross-sectional and 45° tilted SEM micrographs of the scratches on the WT (APS + IM) and DT (APS + IM) samples are also presented in Appendix A. For both samples, there is no visible scratch trace at the early stage as the applied normal force gradually increases from a low value during the nanoscratch test (Appendix A). With further scratching, an apparent scratch trace appears, which becomes wider and deeper with the increasing scratch distance. It is noted, however, that distinct features of the scratch traces are observed for the two samples. For the WT (APS + IM) sample, cracks and tears in the Gr occur around the scratch groove, whereas the DT (APS + IM) sample exhibits a smooth and clean scratch groove without any cracks and tears.

There are two main factors that influence the frictional and wear resistance properties of the Gr/PET films, which also cause different scratch characteristics. One is the structural integrity of the transferred Gr, and the other is the interfacial adhesion between the transferred Gr and PET substrate. As described earlier, the low friction coefficient and excellent mechanical robustness of Gr can effectively improve frictional and wear resistance properties. In this regard, the structural integrity (quality) of the transferred Gr is considered a key factor in improving both properties. As shown in Figure 2, the DT (APS + IM) sample has the most homogeneous and superior structural integrity of the transferred Gr without noticeable structural damages, resulting in a remarkable improvement in both frictional and wear resistance properties.

The interfacial adhesion between the transferred Gr and PET substrate is critical in preventing the delamination of Gr from the PET substrate during the scratch test. When the interfacial adhesion is weak, the Gr coatings around the area in contact with the scratch probe readily delaminate from the PET substrate, making them free-standing. These free-standing Gr coatings are known to fracture prematurely [32]. The delamination-induced premature fracture of the Gr coating brings the scratch tip into direct contact with the PET substrate during the scratch, downgrading the frictional and wear resistance properties of the Gr/PET film (Figure 5a,c and Appendix A). However, the strong interfacial adhesion between the Gr and the PET substrate prevents the delamination-induced premature fracture of the Gr coatings, leading to a smooth and clean scratch trace (Figure 5b,d and Appendix A) and improving its frictional and wear resistance properties.

The delamination of Gr coatings from the PET substrate gives rise to another important effect, i.e., the puckering effect. When a thin sheet such as Gr is scratched, interatomic forces (e.g., van der Waals forces) cause attraction between the sheet and scratch probe causing the sheet to deflect towards the scratch probe and thus increasing the area of interaction between the sheet and scratch probe. This increases friction (friction coefficient). However, the puckering effect will diminish if the sheet strongly adheres to the substrate, thereby suppressing the delamination of the sheet from the substrate [33,34]. In this regard, the enhanced interfacial adhesion between the Gr and the PET substrate achieved by the imidazole treatment restrains the delamination of the Gr from the PET substrate, thereby weakening the puckering effect and improving the frictional properties. This is supported by the experimental results in which the frictional properties were improved by the imidazole treatment for both the wet-transferred and dry-transferred Gr.

## 4. Conclusions

The frictional and wear resistance properties of Gr-coated polymer films (Gr/PET films) and the mechanisms improving both properties were investigated by carrying out nanoscratch tests in combination with scratch trace analysis. The results showed that the desirable combination of Gr’s extremely low friction coefficient and superior mechanical robustness effectively improved both frictional and wear resistance properties. However, the degree of improvement was significantly affected by the fabrication process of the Gr/PET film. The fabrication processes, such as the Gr transfer method and imidazole treatment, influenced two main factors, the structural integrity of the transferred Gr and the transferred Gr/PET interfacial adhesion, which were found to be critical in the improvement of both properties. The dry-transferred Gr possessed a more homogeneous and superior structural integrity than the wet-transferred Gr. The enhanced adhesion of Gr using the imidazole treatment resulted in a strong interfacial adhesion between the Gr and the PET substrate, thereby suppressing the delamination of the Gr from the PET substrate. This enabled the Gr coating to remain on the PET substrate continuously and weakened the puckering effect, which upgraded its frictional and wear resistance properties. Consequently, the dry-transferred Gr with the imidazole treatment provided the best frictional and wear resistance properties for the Gr/PET film.

## Figures and Tables

**Figure 1 nanomaterials-13-00655-f001:**
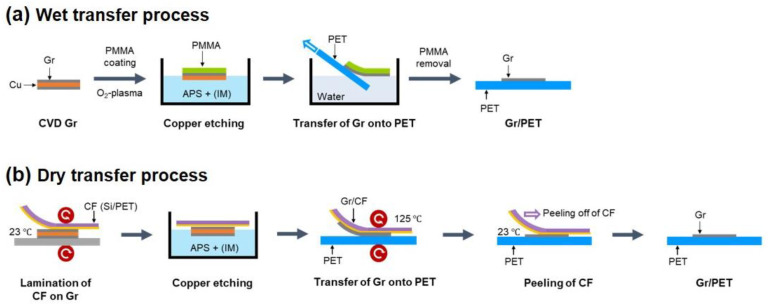
Schematics showing the fabrication processes of the Gr/PET films with two conventional Gr transfer methods. (**a**) Wet transfer and (**b**) dry transfer methods.

**Figure 2 nanomaterials-13-00655-f002:**
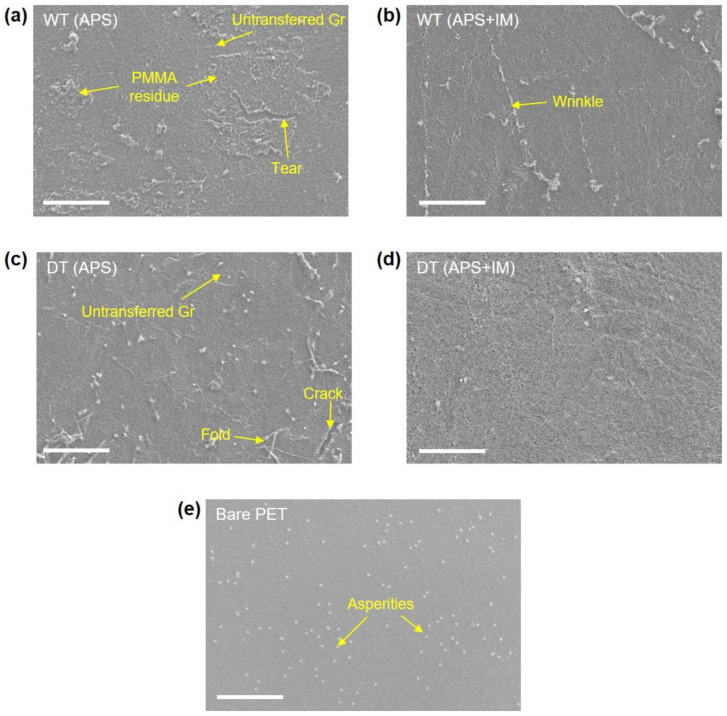
SEM micrographs showing the surface morphology of the Gr/PET films and the bare PET substrate (scale bar: 5 μm). (**a**) WT (APS), (**b**) WT (APS + IM), (**c**) DT (APS), (**d**) DT (APS + IM), and (**e**) bare PET.

**Figure 3 nanomaterials-13-00655-f003:**
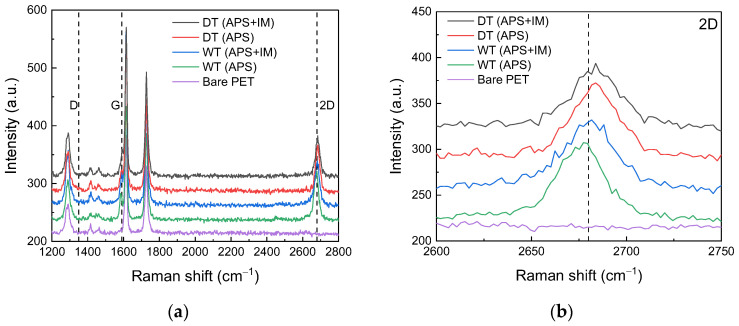
Raman characterization of the microstructural properties of the Gr/PET films and the bare PET substrate. (**a**) Raman spectra and (**b**) enlarged graph comparing Raman 2D bands.

**Figure 4 nanomaterials-13-00655-f004:**
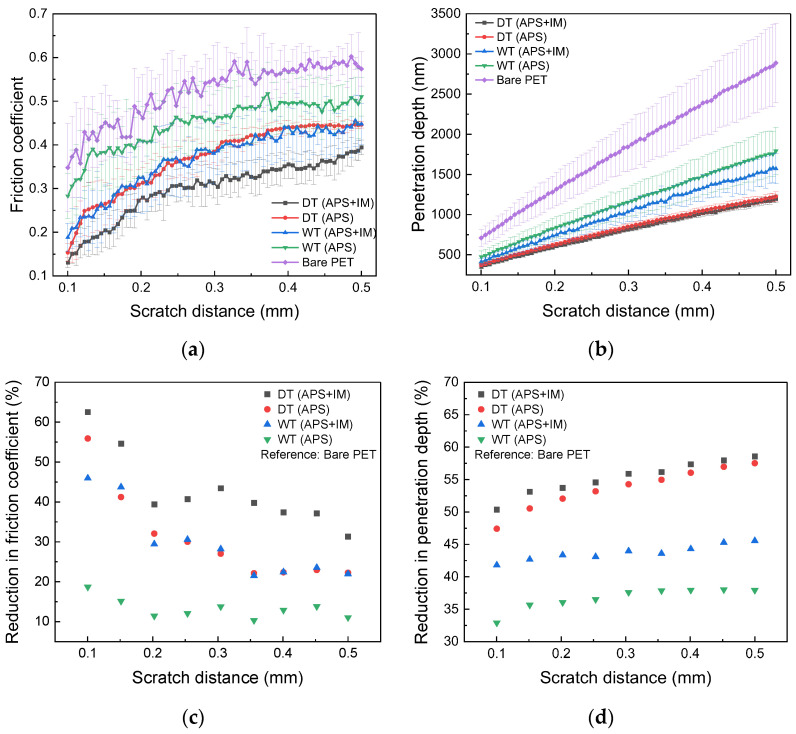
Frictional and wear resistance properties of the Gr/PET films and the bare PET substrate as a function of scratch distance. (**a**) Friction coefficient, (**b**) penetration depth, and reduction ratios in (**c**) the friction coefficient and (**d**) penetration depth with respect to the bare PET.

**Figure 5 nanomaterials-13-00655-f005:**
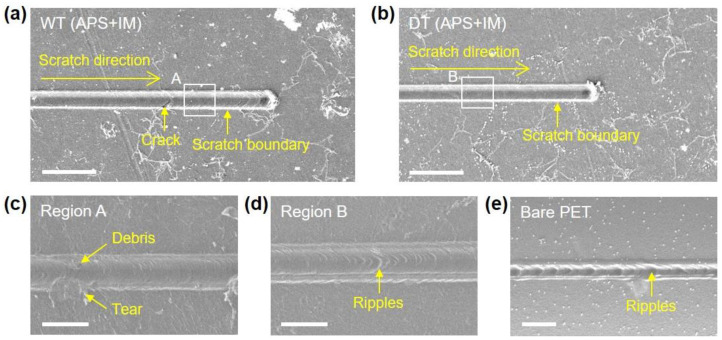
SEM micrographs showing the scratch traces on sample surfaces. (**a**) WT (APS + IM) and (**b**) DT (APS + IM) (scale bar: 20 µm). Enlarged images of (**c**) region A in (**a**), (**d**) region B in (**b**), and (**e**) bare PET substrate (scale bar: 4 µm).

## Data Availability

All data and methods are presented in the main text. Any other relevant data are available from the corresponding author upon reasonable request.

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
