# Peer review of "Effects of the Transfer Method and Interfacial Adhesion on the Frictional and Wear Resistance Properties of a Graphene-Coated Polymer"

_nanomaterials, 2023, doi:10.3390/nano13040655_

Round 1
Reviewer 1 Report
In this paper the authors investigate the frictional and wear resistance properties of graphene deposited on polyethylene terephthalate using two transfer methods (wet and dry) with and without treatment with imidazole. The results suggest that the treatment with imidazole reduces the damage of the graphite during the transfer process and the coating of the polymer with graphite improves its frictional properties. The possible mechanisms leading to the improvement of the frictional and wear resistance properties are discussed. the paper is well-articulated and the discussion and conclusion are well supported by the ex[erimental results presented. BAsed upon te foregoing, I recommend the acceptance of this manuscript in Nanomaterials.
Reviewer 2 Report
The article is devoted to improving the wear resistance of the polymer (PET) by applying a graphene film to the surface. Several methods for applying graphene to the PET surface have been analyzed. The main conclusion is the advantage of the dry transfer method over the wet one, as well as a significant improvement in the wear resistance of the polymer when the surface is treated with imidazole, which leads to better adhesion of the graphene film.
The article is well written and clear. All conclusions are sufficiently substantiated. The reviewer has only a few comments.
1. You propose rather complex procedures for applying graphene to the surface of a polymer. Have you estimated the cost of these procedures and the possibility of their practical application?
2. The colors of the lines in Figure 3 are improper; the lines of similar colors are visually poorly distinguishable.
3. The scale of the y-axis in Figure 5a is chosen poorly, it is impossible to estimate the difference in errors for different systems. In my opinion, it is better to give standard deviations in the text of the article.
4. But in Figure 4c, the errors are absolutely necessary. In their absence, questions arise regarding the extremal nature of the above dependences, especially, for DT (APS+IM).
In general, after minor revision, the article can be accepted for publication.
Reviewer 3 Report
The authors study the influence of graphene coating of a polymer (PET) on the frictional and wear resistance as a function of graphene transfer method and interfacial adhesion.
Although welcome, the results are not unexpected and can be of use for practical applications. However, to be of real interest, the results presented in the manuscript must be complemented by other data.
In particular, to better understand the difference in structural property of Gr-coated PET with respect to the bare PET, a cross-sectional SEM (along some regions of interest) for the samples in fig. 2 is required. The top-view SEM in fig. 2 does not provide the whole information
Same investigations (i.e., cross-sectional SEM) must be provided with respect to fig. 5. From fig. 5 it is not clear at all that "there is no visible scratch trace at the early stage"; in fact, the figure shows the opposite. Then, no meaningful comparison can be made between the WT and DT samples in fig. 5 because it seems that 1) the scratch distance is different, and 2) the regions where enlarged images are provided are situated at different locations/scratching distances from the starting point of testing. In addition, the comparison with the bare PET sample is totally irrelevant since the scale bars are not the same
In summary, without cross-sectional SEMs the manuscript is not convincing
Round 2
Reviewer 3 Report
The manuscript can now be published as is
The authors have responded adequately to the issues raised in the review process